# Fluid Restriction Decreases Solid Food Consumption Post-Exercise

**DOI:** 10.3390/nu11061209

**Published:** 2019-05-28

**Authors:** Cristian Pérez-Luco, Francisco Díaz-Castro, Carlos Jorquera, Rodrigo Troncoso, Hermann Zbinden-Foncea, Neil M Johannsen, Mauricio Castro-Sepulveda

**Affiliations:** 1Carrera Preparador Físico, Centro de Formación Técnica, Santo Tomás, Santiago 8940000, Chile; cperez.afs@gmail.com; 2Laboratorio de Investigación en Nutrición y Actividad Física (LABINAF), Instituto de Nutrición y Tecnología de los Alimentos, Universidad de Chile, Santiago 7810000, Chile; diazexph@gmail.com (F.D.-C.); rtroncoso@inta.uchile.cl (R.T.); 3Nutrition and Exercise Laboratory, Faculty of Medicine, Universidad Mayor, Santiago 8580000, Chile; jorquera@mayor.cl; 4Advanced Center for Chronic Diseases (ACCDiS), Universidad de Chile, Santiago 8380000, Chile; 5Exercise Science Laboratory, School of Kinesiology, Faculty of Medicine, Universidad Finis Terrae, Santiago 7500000, Chile; hzbinden@uft.cl; 6Centro de Salud Deportiva, Clínica Santa Maria, Santiago 7500000, Chile; 7School of Kinesiology, Louisiana State University, Baton Rouge LA 70803, USA; neil.johannsen@pbrc.edu

**Keywords:** energy intake, eating behavior, hormones, appetite, exercise-induced dehydration

## Abstract

Dehydration in rodents induces anorexia. In humans however, the role of dehydration in energy intake is controversial. This study investigated the effect of extreme fluid restriction on acute energy intake during and after exercise-induced dehydration. Eight physically active participants performed two exercise sessions to induce dehydration. After the exercise, the men were allowed to either rehydrate for 2 h or were maintained in a hypohydrated state, in a randomized manner. After 2 h, they were given cereal bars ad libitum for 1 h. Blood and saliva samples of the participants were collected before the exercise session, after the exercise session, after rehydration, and after the meal. Post-exercise energy intake differed between hypohydrated (1430 ± 210 kcal) and rehydrated (2190 ± 780 kcal) trials (*p =* 0.01). For the concentrations of ghrelin and leptin, there were no significant effects of time (*p =* 0.94, *p =* 0.21), between trials (*p =* 0.09, *p* = 0.99), or due to a trial–time interaction (*p =* 0.64, *p =* 0.68), respectively. The concentrations of peptide YY (PYY) were not different between trials (*p =* 0.94) but there was a significant effect of time (*p =* 0.0001) and a trial–time interaction (*p =* 0.01), with higher levels in the rehydration trial after eating in response to a higher energy intake. For saliva production, there was a significant effect of time (*p =* 0.02) and a trial–time interaction (*p =* 0.04), but no between-trial effect (*p =* 0.08). In conclusion, extreme fluid restriction decreased acute food intake after exercise, which may be explained by a lower flow of saliva.

## 1. Introduction

Energy intake depends largely on appetite, which is, in part, regulated by the endocrine system, mainly by the hormones ghrelin, peptide YY (PYY), and leptin. Ghrelin is synthesized by the stomach and is the main orexigenic hormone [1]. PYY and leptin are primary anorexigenic hormones and are released by ileum cells and adipocytes, respectively [1,2]. Anther factor affecting energy intake is water balance. Dehydration in rodents provokes anorexia proportional to the degree of dehydration [3]. This dehydration-anorexia model has been useful in understanding the functional networks that interact to control intake behavior [4].

The role of dehydration in energy intake is not clear in humans. An early study showed that 40% of drinking water limitation is associated with a voluntary reduction in food intake in comparison to a control group [5]. Schirrefs et al. showed that volunteers who were restricted for 37 h from any intake of fluid decreased their overall energy intake by 28% [6]. In contrast, two recent studies showed no effect of dehydration on solid energy intake. Kelly et al. (2012) did not observe changes in post-exercise food intake in participants who started an exercise session in hypohydrated and hydrated states [7]. Corney et al. (2015) showed similar energy intakes in euhydrated and hypohydrated participants the day after an exercise-induced dehydration session [8]. However, in both studies, the participants had access to fluid before or during the meal. The purpose of this study is to determine if extreme fluid restriction after exercise affects the intake of solid food, and if the reduction of solid food intake depends on changes in appetite hormones or the salivary flow rate.

## 2. Materials and Methods

### 2.1. Participants

This study recruited eight physically active college men (age 25.1 ± 3.6 years; body weight 81.1 ± 9.4 kg; body mass index (BMI) 25.6 ± 2.1 kg/m^2^; maximum oxygen consumption (VO_2_max) 52.2 ± 3.3 mL/kg/min^1^). The subjects met the following inclusion criteria: (1) active male volunteers who performed physical activity 3–4 times/week; (2) without musculoskeletal injuries; and (3) who had not consumed coffee or alcoholic drinks 12 h before the study. Only men were recruited, given that the phases of the menstrual cycle in women can modify energy intake [9]. The experimental procedures, associated risks, and benefits were explained to each subject, who then signed a written consent form before participation. The study was developed in accordance with the latest version of the Helsinki Declaration, and was approved by the Ethics Committee of the University Mayor.

### 2.2. Preliminary Testing

Body mass and height were assessed with a high precision balance (0.1 kg) with a built-in stadiometer (SECA model M20812, Hamburg, Germany). The maximal heart rate (MHR) and VO_2_max were determined by an incremental test on a stationary bicycle (OXFORD, model BE 2700, Santiago, Chile) according to the American College of Sports Medicine (ACSM) guidelines for trained participants [10]. VO_2_ was measured during the entire testing period with a metabolic cart (Cosmed FitMate Pro, Rome, Italy).

### 2.3. Experimental Trial

The participants performed two sessions of exercise-induced dehydration with two intervention trials in a randomized, cross-over design, separated by at least one week. Prior to the sessions, urinary specific gravity (USG) was evaluated to assure a euhydrated state (USG < 1.020). Participants were instructed to avoid vigorous physical activity for the day preceding the trials and to maintain their habitual dietary pattern. In addition, alcohol, tobacco, and caffeine-containing drinks were not allowed for the last 12 h before each trial. Participants fasted for 10 h and were given a standard 400 kilocalorie meal (60% carbohydrate, 30% protein, and 10% fat) when they arrived at the laboratory. In the rehydration condition (RH), participants performed 90 min of exercise on a stationary bicycle at an intensity of 60% of maximal heart rate. The heart rate was monitored every 5 min by a telemetric monitor (Polar, RS100, Helsinki, Finland). No fluids were provided during the exercise protocol. Weight loss (loss of fluids) was assessed through the reduction on body mass (kg) calculated by measuring body mass before and after the exercise session (pre-body mass against post-body mass) using the same scale (SECA model M20812, Hamburg, Germany), with a precision of 0.1 kg. After the exercise session, the participants in the rehydration condition (RH) were given 150% of their weight loss (1.5 L for 1 kg of body mass loss) in water to drink [11,12].

Body mass measurements were taken from nude participants after urination. After the rehydration period, each participant received solid cereal bars ad libitum for an hour. Each cereal bar contained 92 kcal, of which 57% (12.9 g) was carbohydrate, 4% protein (0.9 g), and 39% fat (4 g), maintaining the fluid restriction. Participants were instructed to “eat until you are full”. After a week, the participants were subjected to the hypohydrated trial, which was identical to the first except that after exercise, the participants were not allowed to drink any fluids (hypohydration condition; HY) during the 3 h post-exercise period. Blood samples were obtained from an antecubital vein before exercise (BE), after exercise (AE), after rehydration (AE2), one hour after eating the meal (AM), and two hours after eating the meal (AM2) to assess changes in plasma hormones. Samples of saliva were also taken BE, AE, and AE2 to evaluate the saliva flow rate (Figure 1). The protocol lasted approximately eight hours, starting at 10:00 a.m. Participants were asked to consume the same diet before the two trials.

### 2.4. Measurements of Urine Specific Gravity (USG)

The USG was determined with a portable Refractometer (Robinair, model SPX, Michigan, USA) in triplicate according to Castro et al. (2015) [13].

### 2.5. Hormone Measurements

Blood samples for hormone measurements were obtained by cannula in the antecubital vein. Blood samples in heparinized tubes were stored on ice and then centrifuged at 1500 g for 15 min. Plasma was separated and stored at –80 °C until measurements. Concentrations of total ghrelin, total PYY, and leptin were measured in duplicate using commercially available ELISA kits (EMD Millipore Corporation, USA) according to the manufacturer’s instructions. The optical density was determined with a plate reader (Multiskan, Thermo^®^, USA) at 450 nm. Intra-assay and inter-assay variations of the kits were 1.1–1.9% and 5.2–7.7%; 0.9–5.8% and 3.7–16.5%; and 2.6–4.6% and 2.6–6.2%, respectively.

### 2.6. Saliva Measurements

Saliva was collected by the spit method [14] for the measurement of saliva flow rate. Prior to collection, each participant was instructed not to breathe through the mouth or to speak during harvesting. Saliva flow was collected for 1 min for evaluation; each participant spat out the maximum amount of saliva in a disposable container. The volume of saliva was determined by weighing (1.0 mg, equivalent to ~1 mL) on an analytical balance (Radwag AS 220.R2).

### 2.7. Energy Intake

Daily energy intake was quantified by dietary food records, for which participants were previously instructed. Participants were asked to record their food intake the day before the exercise trial (DBE), the day of the exercise trial (DE), and one and two days after the exercise trial (DAE and SDAE, respectively). Energy consumption information was examined using CompEatPro diet analysis software (Nutrition Systems, England, Banbury).

### 2.8. Statistical Analyses

The data are presented as means ± SD. The Shapiro–Wilk test was used to evaluate the normality of the data. A parametric test (Student’s *t*-test) was used to compare acute energy consumption, USG, and weight loss between trials. A two-way repeated measures analysis of variance (ANOVA) was used to compare energy intake before and after the trial day as well as hormone levels, and saliva fluid rate. Statistical significance for multiple comparisons was adjusted using the Bonferroni post-hoc comparison method. An adjusted value of *p* < 0.05 was considered significant. All statistical calculations were performed using the STATISTICA statistical package (Version 8.0; StatSoft Inc., Tulsa, OK, USA).

## 3. Results

Ambient temperature (HY: 30 ± 1 °C; RH: 31 ± 1 °C) and relative humidity (HY: 44 ± 1%; RH: 42 ± 1%) were similar between trials (*p =* 0.91 and *p* = 0.76, respectively) during the dehydration phase. The groups did not present baseline differences in body weight (RH: 82.4 ± 3.7 kg; HY: 80.3 ± 8.1 kg; *p =* 0.62) or BMI (RH: 24.2 ± 5.7 kg/m^2^; HY: 26.3 ± 6.4 kg/m^2^, *p =* 0.28). No difference was found in USG between HY (1.009 ± 0.003) and RH (1.007 ± 0.003) pre-exercise trials (*p =* 0.19). There was no statistical difference in weight lost (loss of fluids) between HY (–2.3 ± 0.3 kg) and RH (–2.2 ± 0.5 kg) at the end of the exercise (*p =* 0.44); the fluid loss percentages were 2.9 ± 0.5% and 2.7 ± 0.6%, respectively (*p =* 0.47). Post-exercise energy intake was different between HY (1430 ± 210 kcal) and RH (2190 ± 780 kcal) (*p =* 0.01, Figure 2). Pre-trial day macronutrient consumption was similar (CHO: *p =* 0.49; protein: *p =* 0.61; fat: *p =* 0.78; see Table 1). Pre-trial and post-trial energy intake did not show any significant difference due to trial (*p =* 0.87) or time (*p =* 0.65), and there was no time–trial interaction (*p =* 0.72) for DBE, DE, DAE and SDAE (Figure 3), including cereal bar intake after exercise.

The concentration of leptin showed no significant effects of time (*p =* 0.94), between trials (*p =* 0.09), or a trial–time interaction (*p =* 0.64) (Figure 4). Ghrelin concentrations showed no significant effect of time (*p =* 0.21), between trials (*p =* 0.99), or a trial–time interaction (*p =* 0.68) (Figure 4). PYY concentration was not different between trials (*p =* 0.94), but there was a significant effect of time (*p =* 0.0001) and a trial–time interaction (*p =* 0.01), with higher levels in the RH condition after eating (AM) (Figure 4).

The rate of saliva production decreased over time (*p =* 0.02) and was almost significantly different between the trials (*p =* 0.08), with a significant interaction found between trial and time (*p =* 0.04). The RH condition produced more saliva at AE2 (*p =* 0.003) (Figure 5).

## 4. Discussion

The results of our study show that extreme fluid restriction decreases acute solid food intake, which may be explained by a lower flow of saliva, but not by changes in hormones that regulate appetite. Our results also suggest that the reduction in solid food intake in the participants observed in the hypohydration trial post-exercise is acute, as no differences were found in total energy intake in the days between trials.

The difference in energy intake between hypohydration and post-exercise rehydration is not explained by a deregulation in the hormonal response, since there were no significant differences in any of the hormones evaluated between trials (groups). The only significant difference we observed was in the time–trial interaction for PYY; specifically, the difference found in the AM time (Figure 4C), where PYY increased in the rehydration condition. This result can be easily explained by the higher energy intake in the rehydration condition where ghrelin concentration is associated with acute energy intake [15].

The flow of saliva seems to be key in regulating post-exercise energy intake, since it was the only measured variable that differed significantly among the trials. Exercise-induced dehydration has been shown to produce acute changes in the production of saliva [16,17], and saliva production may be related to appetite and energy intake [18]. A decrease in saliva flow could reduce lubrication and palatability, making it difficult to swallow and taste the flavors of food [19,20], which could regulate food intake negatively independent of the hormonal response.

Our study is limited by the lack of data concerning other hormones that regulate the appetite, and data including the active forms of PYY and ghrelin, although Corney et al. (2015) found no differences in the acetylation of ghrelin between a dehydrated and a rehydrated group [8] Another limitation of our study was the small sample group (eight subjects) which may have made it harder to detect differences. For a practical application in sport, one of the limitations of this study is ecological validity, since it is unlikely that athletes will have access to food but not liquid after exercise. Similarly, replication studies should consider examining the effect with a choice of a real meal or access to applesauce, ice cream, or watermelon ad libitum, since the cereal bars are dry. Another option to add sport applicability in future studies is to examine (1) the effect of incomplete rehydration; (2) if the effect in hormonal changes and food consumption is related to mild and moderate hydration states; and (3) if similar results can be obtained with non-solid foods, for example, pureed or vitamin-supplemented foods. The implications of these findings in people with obesity must be studied for those seeking to promote weight loss. There is evidence to suggest that drinking immediately before a meal reduces food intake in adults [21]. Therefore, there are likely some settings where rehydration before a meal might impact energy intake negatively. It is possible that the protracted rehydration period in our study prevented this from happening.

## 5. Conclusions

Not consuming fluids during and after exercise decreases acute solid food intake in young physically active adults, which may be explained by a lower flow of saliva.

## Figures and Tables

**Figure 1 nutrients-11-01209-f001:**
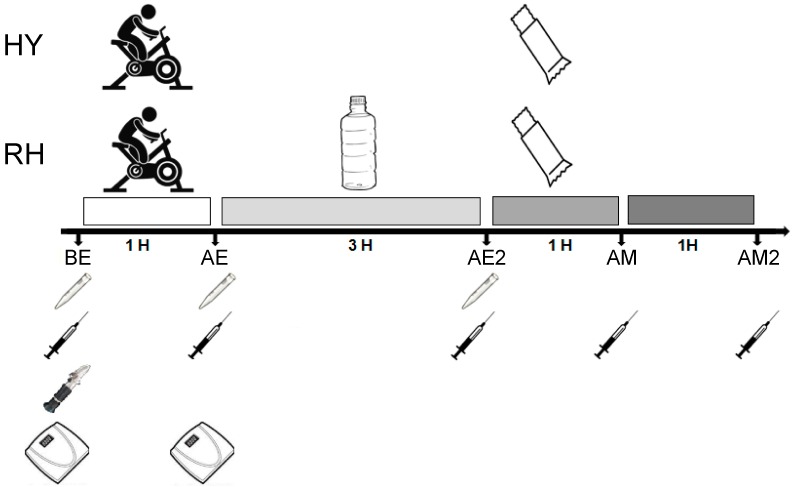
Experimental protocol. BE = pre-exercise; AE = after exercise; AE2 = after hydration trials; AM = after meal; AM2 = 2 h after meal.

**Figure 2 nutrients-11-01209-f002:**
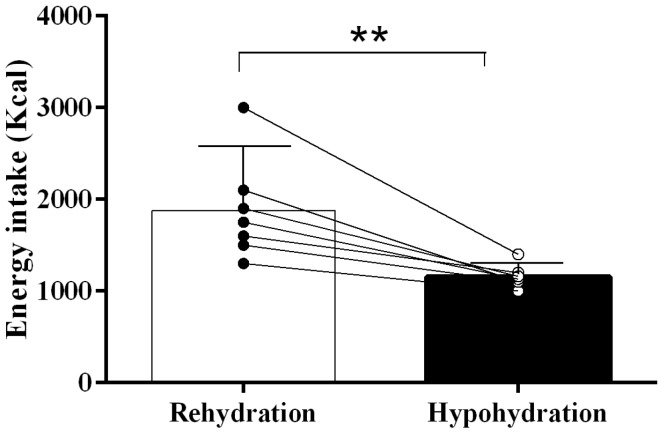
Post-exercise caloric intake. HY = hypohydration condition; RH = rehydration condition; Mean ± SD; ** indicates *p* < 0.01.

**Figure 3 nutrients-11-01209-f003:**
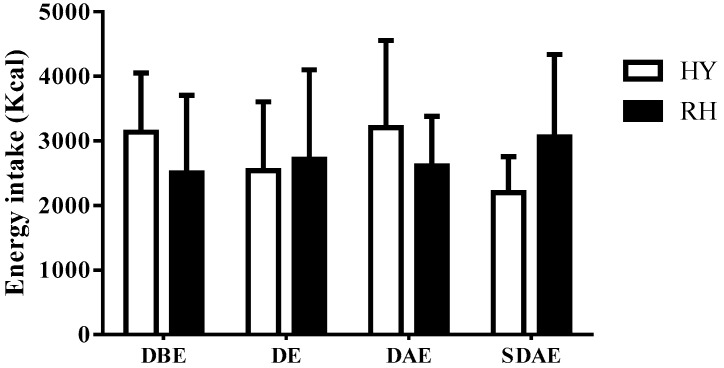
Total calorie intake in hypohydration and rehydration trials before exercise day, and after one and two exercise days. HY = hypohydration condition; RH = rehydration condition; DBE = day before exercise; DE = day of exercise; DAE = day after exercise; SDAE = second day after exercise. Mean ± SD.

**Figure 4 nutrients-11-01209-f004:**
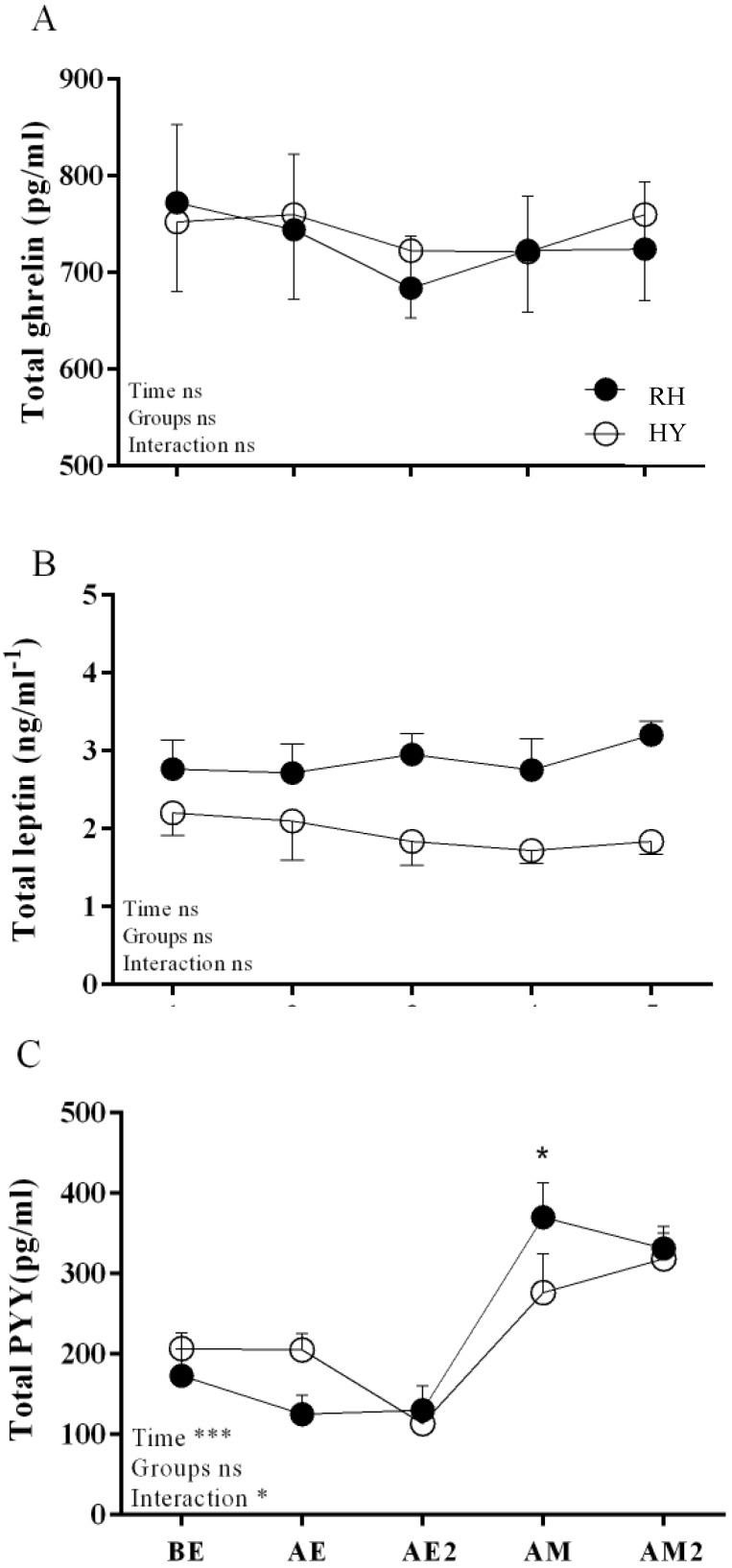
Appetite hormone responses in HY and RH trials. (**A**) Total ghrelin, (**B**) Total leptin, and (**C**) total PYY; HY = hypohydration condition; RH = rehydration condition. BE = pre-exercise; AE = after exercise; AE2 = after hydration trials; AM = after meal; AM2 = 2 h after meal. Mean ± SEM. * indicates *p* < 0.05.

**Figure 5 nutrients-11-01209-f005:**
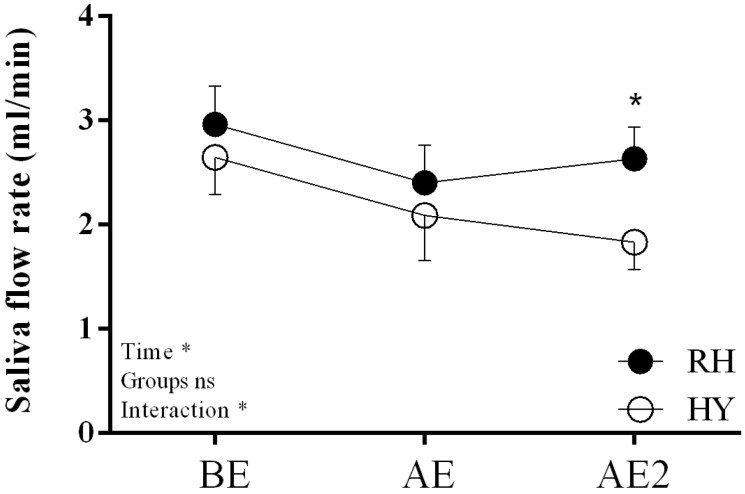
Salivary responses in HY and RH trials; HY = hypohydration condition; RH = rehydration condition; BE = pre-exercise; AE = after exercise; AE2 = after hydration trials. Mean ± SEM; * indicates *p* < 0.05.

**Table 1 nutrients-11-01209-t001:** Macronutrients intake one day before trials.

(%)	RH	HY	*p*-Value
CHO	57 ± 08	55 ± 11	0.49
Protein	31 ± 07	32 ± 08	0.61
Fat	12 ± 04	13 ± 03	0.78

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
