# Peer review of "Fluid Restriction Decreases Solid Food Consumption Post-Exercise"

_nutrients, 2019, doi:10.3390/nu11061209_

Round 1
Reviewer 1 Report
This is a great piece of work concise and relevant, however, the authors need to clarify the following:
There are too many grammatical errors, therefore extensive editing of English Language and style is required
Three protocols were identified; Pre-exercise fast, post-exercise rehydration and fluid abstinence after exercise. Authors need to clarify the protocol used for calculating " weight lost in water", and why 150% of that amount was used to rehydrate participants?
Participants were not allowed to drink any fluids 3 hours post-exercising for 90 minutes at 60% of maximum heart rate. Was the USG measured after 90 minutes or 3 hours post-exercise? Hypohydration can be mild, moderate or severe.
Protocols 2 and 3 will lead to two extremities of hydration state which were not properly explained by the authors. The authors reported that appetite inducing hormonal concentrations at those extremities did not change significantly. Authors may need to investigate food consumption and hormonal changes at mild and moderate states and if similar results will be obtained with non-solid food, for example, pureed or vitamised foods.
Author Response
Comments and Suggestions for Authors
This is a great piece of work concise and relevant, however, the authors need to clarify the following:
· There are too many grammatical errors, therefore extensive editing of English Language and style is required
Our Response
The manuscript was edited by an English Language Editor.
· Three protocols were identified; Pre-exercise fast, post-exercise rehydration and fluid abstinence after exercise. Authors need to clarify the protocol used for calculating "weight lost in water", and why 150% of that amount was used to rehydrate participants?
Our Response
- Body mass loss (kg) was calculated by measuring body mass before and after exercise session (body mass pre – body mass post) using the same scale (SECA model M20812, Germany), with a precision of 0.1 kg.
- The American college of sports medicine recommends ~1.5 L of fluid for each kilogram of body weight lost, for rapid and complete recovery from dehydration. The citation has been added.
11 American College of Sports Medicine, Sawka MN, Burke LM, Eichner ER, Maughan RJ, Montain SJ, Stachenfeld NS. American College of Sports Medicine position stand. Exercise and fluid replacement. Med Sci Sports Exerc. 2007 Feb;39(2):377-90.
· Participants were not allowed to drink any fluids 3 hours post-exercising for 90 minutes at 60% of maximum heart rate. Was the USG measured after 90 minutes or 3 hours post-exercise? Hypohydration can be mild, moderate or severe.
Our Response
We do not evaluate USG during the protocol because USG is not considered a good marker of acute changes in hydration state during exercise 1, 2.
1 Popowski LA, Oppliger RA, Patrick Lambert G, Johnson RF, Kim Johnson A, Gisolf CV. Blood and urinary measures of hydration status during progressive acute dehydration. Med Sci Sports Exerc. 2001 May;33(5):747-53.
2 Hamouti N, Del Coso J, Mora-Rodriguez R. Comparison between blood and urinary fluid balance indices during dehydrating exercise and the subsequent hypohydration when fluid is not restored. Eur J Appl Physiol. 2013 Mar;113(3):611-20.
· Protocols 2 and 3 will lead to two extremities of hydration state which were not properly explained by the authors. The authors reported that appetite inducing hormonal concentrations at those extremities did not change significantly. Authors may need to investigate food consumption and hormonal changes at mild and moderate states and if similar results will be obtained with non-solid food, for example, pureed or vitamised foods.
Response
This is a limitation of our study, we only had two experimental group to induce an extreme dehydration state, one was the control group with access to water before the meal and the other group private the any water consumption before or during the meal. We appreciate his comment and believe that are an interest topic for futures investigation. We will add this to the limitation section.

Reviewer 2 Report
Dear Authors,
You have taken up an interesting topic, however, for better understanding, it is worth completing the manuscript with the following elements:
Introduction:
lines 58-59: Information on stude results should not be included in the introduction. I would like to know more about the purpose of the work; especially what is different from the presented approach from other studies.
Participants:
What were the inclusion and exclusion criteria for study participants? The average value of BMI suggests that not all had normal weight (the state of nutrition was not uniform in the group?)
Experimental trial:
The study protocol is not fully clear for me. Presentation of the research scheme in the form of a scheme would increase readability for the reader.
Discussion:
To the limitation of the study I would add the small sample group (8 people).
General remark:
It seems to me that the final conclusion is too general. The study was conducted only on young men, so the conclusions should apply only to this group, not general population.
The study protocol compares two situations: 1. very unusual / not physiological (not drinking during and after physical exercise) 2. typical (recommended hydration after physical exertion). In order to obtain more valuable results, it is worth considering performing the test with incomplete hydration after exercise (which happens in practice), as well as with excessive fluid intake after exercise. Then the results / conclusions could have greater practical application.
Author Response
Comments and Suggestions for Authors
Dear Authors,
You have taken up an interesting topic, however, for better understanding, it is worth completing the manuscript with the following elements:
Introduction:
lines 58-59: Information on stude results should not be included in the introduction. I would like to know more about the purpose of the work; especially what is different from the presented approach from other studies.
Our Response
As suggestions of the reviewer we added the purpose of the study before the summarize the principal findings of the work.
Participants:
What were the inclusion and exclusion criteria for study participants? The average value of BMI suggests that not all had normal weight (the state of nutrition was not uniform in the group?)
Our Response
“The subjects met the following inclusion criteria: 1) Active men, volunteers performed physical activity 3-4 times / week. 2) Do not present musculoskeletal injuries. 3) Do not have consumed coffee or alcoholic drinks 48 hours before the study”. This sentence has been added to methods section.
“The groups did not present baseline differences in body weight (RH; 82.4±3.7, HY; 80.3±8.1 kg, p=0.62); neither BMI (RH; 24.2±5.7, HY; 26.3±6.4 Kg/m2, p=0.28)”. This sentence has been added to results section.
Experimental trial:
The study protocol is not fully clear for me. Presentation of the research scheme in the form of a scheme would increase readability for the reader.
Response
A scheme has been added
Discussion:
To the limitation of the study I would add the small sample group (8 people).
Our Response
The comment has been added to limitation of the study
General remark:
It seems to me that the final conclusion is too general. The study was conducted only on young men, so the conclusions should apply only to this group, not general population.
Our Response
The conclusion was bounded to young physically active adults.
The study protocol compares two situations: 1. very unusual / not physiological (not drinking during and after physical exercise) 2. typical (recommended hydration after physical exertion). In order to obtain more valuable results, it is worth considering performing the test with incomplete hydration after exercise (which happens in practice), as well as with excessive fluid intake after exercise. Then the results / conclusions could have greater practical application.
Our response
This is a limitation of our study, we only had two experimental group to induce an extreme dehydration state, one was the control group with access to water before the meal and the other group private the any water consumption before or during the meal. We appreciate his comment and believe that are an interest topic for futures investigation. We will add this to the limitation section.

Round 2
Reviewer 1 Report
Authors have not properly addressed:
"weight lost in water"??
Authors cited "~1.5 L of fluid for each kilogram of body weight lost, for rapid and complete recovery from dehydration". -Good.
How many kilograms of body mass was lost by the participants if 150% of this amount was given to each for rehydration? Please find out the mass of 1 litre of water in kg.
Authors should rewrite line 52 and delete "and during the meal "
Author Response
Comments and Suggestions for Authors
Authors have not properly addressed:
· "weight lost in water"??
Our response
This was better explained
· Authors cited "~1.5 L of fluid for each kilogram of body weight lost, for rapid and complete recovery from dehydration". -Good.
Our response
Thanks for your comment.
· How many kilograms of body mass was lost by the participants if 150% of this amount was given to each for rehydration? Please find out the mass of 1 litre of water in kg.
Our response
The kilograms in the reduction of body mass lost are show between the lines 155 and 157.
In the section 3.2 “Experimental trial” we added the explication “(1.5L for 1Kg of body mass loss)”.
· Authors should rewrite line 52 and delete "and during the meal "
Our response
The line 52 was corrected.
Reviewer 2 Report
I appreciate the authors' work in introducing the suggested changes.
I believe that the article gained substantive after the introduction of changes.
Author Response
Thanks for your comment.